# Knowledge Graph Recommendation Model Based on Feature Space Fusion

**Suqi Zhang** [1,*], **Xinxin Wang** [2], **Rui Wang** [3], **Junhua Gu** [3,4] **and Jianxin Li** [5]

1   School of Information Engineering, Tianjin University of Commerce, Tianjin 300134, China
2   School of Science, Tianjin University of Commerce, Tianjin 300134, China
3   School of Artificial Intelligence and Data Science, Hebei University of Technology, Tianjin 300401, China
4   Hebei Province Key Laboratory of Big Data Computing, Tianjin 300401, China
5   School of IT, Deakin University, Burwood, VIC 3125, Australia
*   Correspondence: zhangsuqie@163.com

**Abstract:** The existing recommendation model based on a knowledge graph simply integrates the behavior features in a user–item bipartite graph and the content features in a knowledge graph. However, the difference between the two feature spaces is ignored. To solve this problem, this paper presents a new recommendation model named the knowledge graph recommendation model based on feature space fusion (KGRFSF). Specifically, in the behavioral feature space, the behavioral features of users and items are constructed by extracting the behavioral feature from the user–item bipartite graph. In the content feature space, the content features related to users and items are extracted through the attention mechanism on the knowledge graph, and then the content feature vectors of users and items are constructed. Finally, through the feature space fusion model, the behavior features and content features are projected into the same preference feature space, and then the fusion of the two feature spaces is completed to construct the complete vector representations of users and items and calculate the vector similarity to predict the score of the user to the item. This paper applies the presented model to public datasets in the fields of music and film. It can be found through the experimental results that KGRFSF can effectively improve the recommendation accuracy compared with the existing models.

**Keywords:** recommendation system; knowledge graph; collaborative filtering; attention mechanism

## 1. Introduction

Currently, along with the rapid development of the Internet and big data technology, users are facing the question of information overburden. To alleviate information overburden, recommendation systems bring into play an irreplaceable function in applications that provide information services, such as e-commerce, social platforms, and news media [1]. Among the existing recommendation models, the collaborative filter (CF) recommendation model [2–5] is the most widely used model in the recommendation model, which holds that the user's interactive items express the user's direct preference information; the interactive users of the items express the feature information of the items. Based on this, the collaborative filtering recommendation model is organized into item-based and user-based. Item-based methods calculate the similarity matrix between items to measure the similarity between the item and the items in the user interaction history to estimate the user's preference for the item; user-based methods count the similarity matrix between users to find users similar to the current user and then recommend according to the interaction history of similar users. Although the recommendation model based on CF has achieved great success in a large number of application scenarios, it still faces the questions of sparse data [6–8] and cold start [9–12]. The data sparsity refers to the fact that user and item interaction information is often very scarce. For example, in movie recommendation, there are often thousands of movies, but users often rate only a few dozen movies. Using

so little data to predict a large amount of unknown information greatly increases the risk of overfitting. The cold start means that there is no corresponding historical information for newly added users or items, so it is difficult to make accurate recommendations.

A knowledge graph (KG) includes abundant item attribute information and correlation information; among them, the basic structure of the knowledge graph is a directed heterogeneous graph. The nodes in the graph correspond to the entities, and the edges correspond to the semantic relationships between entities. Further, $(h, r, t)$ represents the knowledge graph triple; $h \in \varepsilon, r \in R, t \in \varepsilon$ represent the head entity, relation, and tail entity in the triple, respectively; $\varepsilon$ and R represent the set of entities and relations in the knowledge graph, respectively; and the knowledge graph is brought into the recommendation system as assistant information, which can alleviate the questions of sparse data and cold start. Therefore, recommendation models based on knowledge graphs have received increasing attention from researchers. Wang et al. proposed the Ripple Net [13], which takes the item as the center, spreads outward along the relationship in the KG to aggregate the information of the surrounding nodes, and then constructs the feature vectors of users and items. Wang et al. proposed the KGCN [14], which introduces a graph convolutional network into the knowledge graph recommendation field. KGCN combines the entity node's own information and neighborhood node information to calculate the entity embedding vector containing high-order correlation information. Wang et al. proposed the KGAT [15], which integrates the user–item bipartite graph (UIG) and the knowledge graph (KG) into a collaborative knowledge graph (CKG). KGAT assembles the neighborhood information of nodes through the attention mechanism on the CKG and excavates the high-order correlation information between entities by stacking multilayer network structures to construct the embedded vector representations of users and items. Wang Z. et al. proposed the CKAN [16], which obtains the feature vector of the item through the attention mechanism embedded in the content feature in the KG. The user is then represented as the sum of all item feature vectors in their interaction history; according to the correlation between the user vector and the item vector, the score of the user to the item is predicted.

In summary, many current recommendation models based on knowledge graphs combine UIG and KG into CKG and are based on graph convolutional networks for information propagation on CKG, which is equivalent to using the content in the knowledge graph to explain all the interaction behaviors of users. However, in real scenarios, because the content information contained in the knowledge graph cannot cover all features of the item and the user interaction with the item is not only for item content preferences, the unification of the behavior feature space and content feature space in this kind of model will cause the constructed user features and item features to be mixed with noise, which affects the recommendation accuracy.

Finally, the works of this paper are as follows:

- This paper presents a recommendation model named the knowledge graph recommendation model based on feature space fusion (KGRFSF), which can combine the content feature in KG with the behavior feature in UIG.
- This paper applies the presented model to public datasets in the fields of music and film. The experimental results show that KGRFSF can effectively improve the recommendation performance compared with the existing models.

## 2. Related Works

The recommendation model of knowledge graph [17–21] combines the attribute information of users and items learned from KG on the basis of the collaborative filter recommendation model, calculates the vector representation of the user and the item, and then evaluates the user's preference for the item by means of the vector internal product. The study of this type of method focuses on how to combine attribute information from the KG into user vectors and item vectors. In recent years, inspired by convolutional networks, methods of graph-based information dissemination have developed rapidly. KGCN [14] samples each neighborhood node in the KG, calculates the weights of the

neighbors based on the relationship between the nodes, and finally combines the neighborhood node information to the center node according to the weights of the neighbors. KGAT [15] combines the UIG and the KG to construct CKG and then applies the attention mechanism to combine the neighborhood node information of the user and the item on a CKG. This type of approach integrates the global information of the KG to enrich the vector representations of users and items. CKAN [16] obtains the feature vector of the item through the attention mechanism embedded in the knowledge graph content feature. The user is then represented as the sum of all item feature vectors in their interaction history. According to the correlation between the user vector and the item vector, the score of the user to the item is predicted. Although optimization of entity embedding by graph convolution can effectively introduce the information in CKG into the recommendation task, at the same time, because the semantic information behind behavioral characteristics and content characteristics is not equal, behavioral features reflect a variety of complex features, such as item content and popularity. Content features are simply features inherent to the item. The method of message propagation on CKG ignores the difference in semantic information between KG and UIG, which results in a noise problem. This is also the focus of the model in this paper.

## 3. Our Model

### 3.1. Problem Analysis and Solution

The item-related content feature contained in the knowledge graph is used to assist the recommendation, which effectively alleviates the questions of sparse data and cold start faced by the collaborative filter recommendation model. However, many current recommendation models based on knowledge graphs combine UIG and KG into CKG and use the graph convolution neural network algorithm to spread information on the CKG. This kind of model simply integrates the behavioral features in the UIG, and the content features in the KG in the same space will ignore that they are actually in different feature spaces and express different semantics. At the same time, the constructed user features and item features are mixed with irrelevant noise information.

For example, as shown in Figure 1, The UIG of the green rectangular and the item-related KG of the blue rectangle constitute CKG. In the UIG, users A and B jointly interact with "The Shawshank Redemption", and they are users with similar behaviors; "The Shawshank Redemption" and "Titanic" have both been interacted with by User B, and the two are movies with similar interaction behavior. After analyzing the behavior features of user groups, it is found that users often choose to watch "Titanic", which is also an Oscar-winning film, after watching "The Shawshank Redemption". Therefore, "Titanic" is likely to be a movie that User A will watch through the message propagation path of the red dotted arrow. In the knowledge graph, the movie that User A has interacted with contains the attributes of "comedy", "plot", and "crime" through the blue dotted arrow, indicating that User A is interested in these contents. Similarly, User B is interested in including "plot," "crime," and "love." When the information is disseminated on the CKG, the information of "love" and "tragedy" is transmitted to "The Shawshank Redemption" and User A through the path indicated by the red dotted arrow in the figure. Although "The Shawshank Redemption" and "Titanic" have been interacted with by User B, the two have no connection in content, and the interaction history of User A does not include movies with content such as "love" and "tragedy". This raises two problems: (1) for the user, because User A is spread with the message of "tragedy", when recommending a movie for User A, the recommendation result will appear in the recommendation result that User A has no interest in "Romeo and Juliet"; (2) for the item, because "The Shawshank Redemption" is spread with "love" and "tragedy" information, "The Shawshank Redemption" will be recommended to users who like "love" or "tragedy".

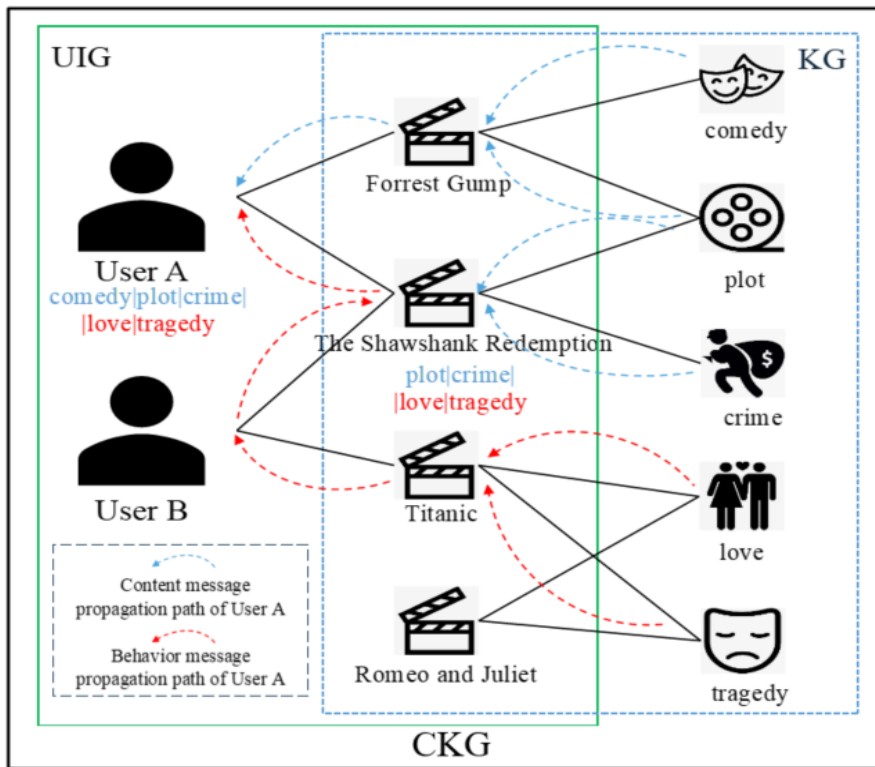

**Figure 1.** Message propagation on CKG.

In fact, the behavioral feature in the UIG and the content feature in the KG contain different semantic information, and the two describe the features of users and items from different aspects. In Figure 1, User A interacts with "Forrest Gump" and "The Shawshank Redemption", and the semantic messages behind the interaction include the following: the two movies are in line with User A's preferences; the two movies are popular; User A was recommended by a friend to watch the two movies; and so on. Therefore, the semantic information behind the user's interaction behavior is complex and diverse. In contrast, the content feature contained in the knowledge graph is more specific: "comedy", "plot", and "crime" information can be interpreted as User A's preference for the specific content of the movie. Through comparison, it can be found that the semantic information contained behind the behavioral feature and content feature is unequal, the behavioral feature reflects the complex factors behind the user interaction, and the content feature reflects the specific content that the user likes. Message propagation on CKG ignores the difference in semantic information between KG and UIG, resulting in the above noise problem.

To solve the abovementioned noise problem, a knowledge graph recommendation model based on feature space fusion (KGRFSF) is proposed in this paper. Next, the feature modeling process of the user is taken as an example to explain the thinking of the feature space fusion method in the model, as shown in Figure 2, considering the semantic differences between behavioral features and content features. First, the behavioral and content features of users are modeled in the behavioral and content feature spaces. In the UIG, the behavioral feature vector $u_{cf}$ of the user is extracted in the behavioral feature space. According to the interaction history of the user, the content entities associated with the interactive items of the user are screened out from KG. In the content feature space, this entity information is combined to form the user's content feature vector $u_{kg}$; then, before the two are fused to output a complete user vector, the behavioral feature and content feature are projected into the preference feature space by the behavioral feature projection matrix and content feature projection matrix $M_{cf}$ and $M_{kg}$, respectively, which simplifies the semantic information behind the behavioral feature and content feature so that the feature vector after the projection only expresses the user's preference information, which is

convenient for subsequent fusion of the two. Finally, the two features are fused in the same feature space by splicing and a multilayer perceptron to obtain the user's complete feature vector $u_{final}$, which avoids the noise problem caused by the simple fusion of behavioral features and content features in the same space.

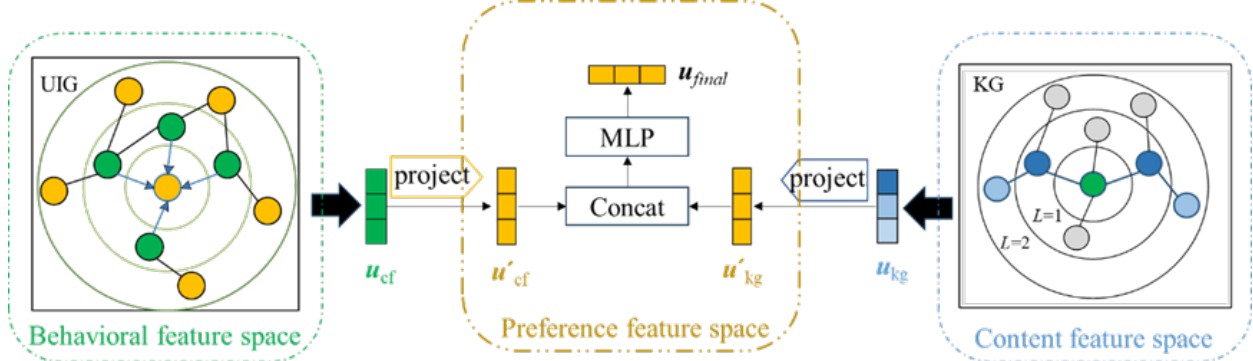

**Figure 2.** User's feature modeling process.

### 3.2. Model Introduction

We present a knowledge graph recommendation model based on feature space fusion (KGRFSF), as shown in Figure 3. There are four modules in the architecture of the recommendation model: (1) Behavioral feature module: for the input user $u$ and item $v$ to be recommended, the corresponding behavioral features are extracted from the UIG to construct the user's behavioral feature vector $u_{cf}$ and item's behavioral feature vector $v_{cf}$. (2) The content feature module: based on the attention mechanism, aggregates the $L$-order neighborhood information of the user interactive item set $\{v_1, v_2, \ldots, v_n\}$ and item $v$ in KG, and the content feature vectors $u_{kg}$ and $v_{kg}$ of users and items are obtained. (3) The feature space fusion module: projects the behavioral feature vectors ($u_{cf}$ and $v_{cf}$) and the content feature vectors ($u_{kg}$ and $v_{kg}$) obtained in the first two steps into the preference feature space through a projection operation. After the concatenation operation, it is input into the multilayer perceptron (MLP) to complete the fusion and obtain the final vector of the user, and the item represents $u_f$ and $v_f$. (4) Prediction module: the vector inner product of the two is calculated as the user's predicted click-through rate $\hat{y}(u, v)$. The complete training process is shown in Algorithm 1.

---

**Algorithm 1:** Knowledge graph recommendation model based on feature space fusion

---

Input: user $u$, item $v$, user–item bipartite graph $G_1 = \{U, I\}$, knowledge graph $G_2 = \{V, E\}$, embedding dimension $d$, learning rate $\eta$, user entity set size $S_u$, item entity set size $S_v$.
Output: score of the user on the item $\hat{y}(u, v)$.
**Step 1: for** $n = 0$ to *epoch* **do**.
**Step 2:** According to Formulas (5) and (6), user behavioral feature embedding $u_{cf}$ and item behavioral feature embedding $v_{cf}$ are calculated.
**Step 3:** User content feature embedding $u_{kg}$ and item content feature embedding $v_{kg}$ are counted according to Formulas (9)–(18).
**Step 4:** According to Formula (19)–(25), calculate the score $\hat{y}(u, v)$ of users.
**Step 5:** Calculation Formula (26).
**Step 6:** Calculate gradient and send back to update model parameters.
**Step 7: end for**
**Step 8: return** $\hat{y}(u, v)$

---

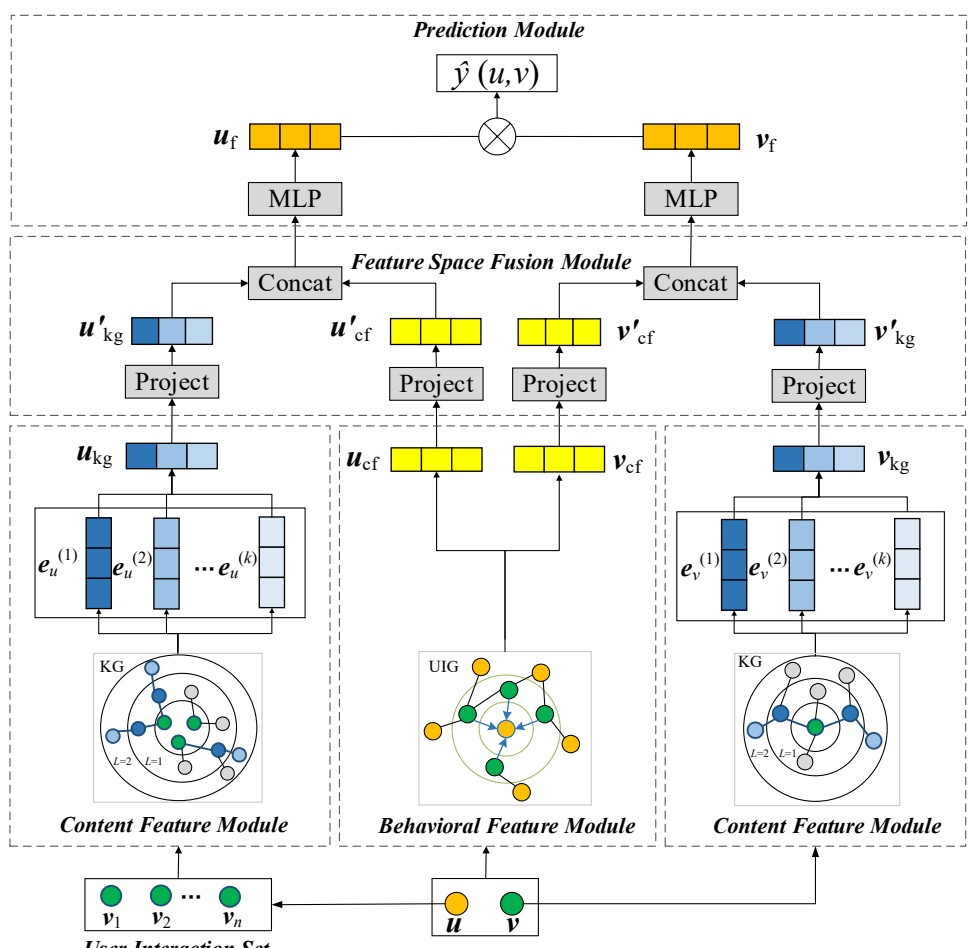

**Figure 3.** Illustration of the proposed KGRFSF model.

### 3.2.1. Behavioral Feature Module

The model extracts the behavioral feature contained in the UIG through the behavioral feature module. Behavioral features are potential factors that influence users' decisions, as reflected by interaction history. For example, when a user watches a movie, in addition to being interested in the content of the movie, they may also be affected by factors such as conformity psychology (many users have watched the movie), publicity (attracted by the advertisements and posters of the film), friend recommendation, etc. The role of these factors is reflected through the user's interaction history, which depicts the user's behavior features. The behavior feature from the UIG and the content feature contained in the KG affect users' decisions from different aspects. As shown in Figure 1, User A interacts with "Forrest Gump" and "The Shawshank Redemption" because both belong to "feature films" in terms of content. User B's interactions "Titanic" and "The Shawshank Redemption" are not due to their similar content feature in the knowledge graph but because they are both highly acclaimed and highly interactive "high score movies" with a large number of users. Users who have watched one of the two movies tend to watch the other movie. Therefore, "Titanic" is probably the movie that User A will watch. In summary, the behavior feature module is an important aspect of modeling users and items.

In the UIG, the interactive items of users express the features of users, and the interactive users of items express the features of items [3]. As a result, the behavioral feature module aggregates the neighborhood information of nodes in the UIG by a graph convolution operation to obtain the behavioral feature vector representation of user $u_{cf}$ and item $v_{cf}$:

$$u^{(k+1)} = \sum_{v \in N_u} \frac{1}{\sqrt{|N_u|}\sqrt{|N_v|}} v^{(k)} \tag{1}$$

$$v^{(k+1)} = \sum_{u \in N_v} \frac{1}{\sqrt{|N_v|} \sqrt{|N_u|}} u^{(k)} \tag{2}$$

where $u^{(k)}$ and $v^{(k)}$ are the vector representations of user $u$ and item $v$ after convolution at layer $k$, $N_u$ is all items for user $u$ interaction, and $N_v$ is all users for item $v$ interaction.

After $K$ layer convolution, $K$ vectors are obtained, and each vector contains neighborhood information of different ranges. The larger $K$ is, the wider the neighborhood range is, and the less information contained is associated with the central node. The model integrates the results of the $k$-layer convolution to obtain the behavioral feature vectors of user $u_{cf}$ and item $v_{cf}$:

$$u_{\text{cf}} = \sum_{k=0}^{K} \alpha_k u^{(k)} \tag{3}$$

$$v_{\text{cf}} = \sum_{k=0}^{K} \alpha_k v^{(k)} \tag{4}$$

where $\alpha_k$ is the weight of the $k$ layer vector, with a value of $1/(k+1)$. For the central node, the closer the node is, the more important its information and the higher its weight.

To understand graph convolution from an overall perspective and facilitate batch processing, the matrix form of each layer of convolution is given here:

$$E^{(k+1)} = \left( D^{-\frac{1}{2}} A D^{-\frac{1}{2}} \right) E^{(k)} \tag{5}$$

where $A$ is the adjacency matrix of nodes in the UIG, $D$ is the degree matrix of nodes in the UIG, and $E$ is the embedding vector matrix of nodes in the UIG. The calculation formula of the output feature matrix of the comprehensive $K$-layer convolution is as follows:

$$E = \alpha_0 E^{(0)} + \alpha_1 E^{(1)} + \alpha_2 E^{(2)} + \ldots + \alpha_K E^{(K)} \tag{6}$$

where $\alpha_K$ represents the weight of the $K$ layer vector, and the value is $1/(K+1)$.

### 3.2.2. Content Feature Module

The behavioral feature extracted from the UIG expresses the potential factors reflected from the interaction behavior that affects the user's decision, while the content feature contained in the knowledge graph specifically describes what content feature users prefer. For users, the associated entities of user interaction items in the knowledge graph reflect the content they are interested in. For example, User A interacts with "Forrest Gump" because User A likes to watch feature films. For items, the neighborhood entities in the KG reflect the content feature of items. For example, "Forrest Gump" is a comedy or drama film. To mine the specific content features of users and items from the KG and enrich the feature expressions of users and items, the model sampled the neighborhood of interactive items of users in the knowledge graph and constructed the user's initial set of content entities. The item's initial set of content entities consists of the item itself. The initial set of content entities is defined as follows:

$$\varepsilon_u^{(0)} = \{v | v \in G \text{ and } y_{uv} = 1\} \tag{7}$$

$$\varepsilon_v^{(0)} = \{v | v \in G\} \tag{8}$$

where $G$ is the set of all entities and relationships in the KG, $y_{uv}$ represents whether the user interacts with the item, a value of 1 indicates interaction, and a value of 0 indicates no interaction. $\varepsilon_u^{(0)}$ is the initial set of content entities for users, and $\varepsilon_v^{(0)}$ is the initial set of content entities for items. Then, starting from the initial set of content entities, the neighbors

of the entities in the set are sampled layer by layer to obtain the relevant content entities of user $u$ and item $v$ in the knowledge graph. We obtain the entity set sampled at layer $l$:

$$\varepsilon_u^{(l)} = \left\{ t \Big| (h, r, t) \in G \text{ and } h \in \varepsilon_u^{(l-1)} \right\} \tag{9}$$

$$\varepsilon_v^{(l)} = \left\{ t \Big| (h, r, t) \in G \text{ and } h \in \varepsilon_v^{(l-1)} \right\} \tag{10}$$

where, in Formula (9), $h$ represents user $u$ as the central node, and the entity set is sampled at layer $l - 1$. in Formula (10), $h$ represents the entity centered on item $v$, and the entity set is sampled at layer $l - 1$.

Next, the content feature related to users and items is aggregated. In the KG, layer by layer, entities connected by different relationships have different degrees of importance to users (or items). For example, User A watches "Forrest Gump" because User A likes to watch feature films. Therefore, when constructing User A's content feature vector, the weight of "feature films" will be higher. To describe this relationship, the model makes the graph attention mechanism count the weight of different entities and then aggregates the content feature in the KG according to the weight.

It is assumed that the triplet $(h, r, t)$ is a triplet of an entity $t$ in the entity set at layer $l$ of user $u$, and the vector of $t$ with added attention weight is defined as $a_t$:

$$a_t = \pi \left( e^h, e^r \right) e^t \tag{11}$$

where $e^h$, $e^r$, and $e^t$ are the vector representations of head node $h$, relation $r$, and tail node $t$, respectively, and $\pi \left( e^h, e^r \right)$ is the normalized attention score function of $t$:

$$\pi \left( e^h, e^r \right) = \frac{\exp \left( \pi' \left( e^h, e^r \right) \right)}{\sum_{t' \in \varepsilon_u^{(l)}} \exp \left( \pi' \left( e^h, e^{r'} \right) \right)} \tag{12}$$

where $\pi' \left( e^h, e^r \right)$ is the attention score function of $t$:

$$\pi' \left( e^h, e^r \right) = \sigma(W_2 ReLU(W_1 z_0 + b_1) + b_2) \tag{13}$$

$$z_0 = ReLU(W_0 \left( e^h || e^r \right) + b_0) \tag{14}$$

where the attention network in the first two layers of the nonlinear activation function is the linear rectifier function (rectified linear unit, ReLU) [22], the last layer activation function is the sigmoid function [23], | | is the concatenation operation, $W$ and $b$ are the learning parameters, and $r'$ represents the relationship corresponding to attribute entity $t'$.

Then, all content features of layer $l$ are aggregated based on the weight to obtain the content vector representation of layer $l$ of user $e_u^{(l)}$:

$$e_u^{(l)} = \sum_{t \in \varepsilon_u^{(l)}} a_t \tag{15}$$

By the same method to count the $l$-level content vector representation of items $e_v^{(l)}$:

$$e_v^{(l)} = \sum_{t \in \varepsilon_v^{(l)}} a_t \tag{16}$$

Finally, the content vector of layer $L$ is integrated to output the content feature vectors of users and items $u_{kg}$ and $v_{kg}$:

$$u_{kg} = \left( e_u^{(1)} || e_u^{(2)} \dots || e_u^{(L)} \right) \tag{17}$$

$$v_{kg} = (e_v^{(1)} || e_v^{(2)} \ldots || e_v^{(L)}) \tag{18}$$

where $||$ denotes the concatenation operation.

### 3.2.3. Feature Space Fusion Module

Behavioral features and content features describe the features of users and items from two different aspects of interactive behavior and knowledge graph content. The fusion of the two features is able to enrich the expressions of users and items and enhance the accuracy of recommendation. However, as analyzed in Section 3.1, behavioral features and content features contain different semantic information and belong to different feature spaces. Direct fusion will result in a "noise" problem. To solve this problem, the model proposes the feature space fusion method: first, in the behavioral feature space, the behavioral features of users and items are constructed by extracting the behavioral feature from the user–item bipartite graph. Then, in the content feature space, the attention mechanism is applied to extract the content information related to users and items from the knowledge graph, and the content feature vector of users and items is constructed. Finally, through the feature space fusion model, on the one hand, through the projection operation, the behavior features and content features are projected into the preference feature space so that they are in the same feature space. Specifically, the behavior feature vector $u_{cf}, v_{cf}$ and content feature vector $u_{kg}, v_{kg}$ are projected into the same preference feature space through the projection matrix $M_{cf}, M_{kg}$. On the other hand, through feature cross, the behavioral feature and content feature extract important feature subset and filter irrelevant semantic features. Specifically, behavioral features and content features are spliced in the preference feature space, and then feature cross is carried out through multilayer perceptron (MLP) to output the complete representations of user and item feature vectors $u_f$ and $v_f$:

$$u'_{cf} = M_{cf} u_{cf} \tag{19}$$

$$u'_{kg} = M_{kg} u_{kg} \tag{20}$$

$$v'_{cf} = M_{cf} v_{cf} \tag{21}$$

$$v'_{kg} = M_{kg} v_{kg} \tag{22}$$

$$u_f = \sigma\left(W\left(u'_{cf} || u'_{kg}\right) + b\right) \tag{23}$$

$$v_f = \sigma\left(W\left(v'_{cf} || v'_{kg}\right) + b\right) \tag{24}$$

where $M_{cf}$ and $M_{kg}$ represent the behavioral feature projection matrix and content feature projection matrix, respectively, and $W$ and $b$ are the parameters to be trained.

### 3.2.4. Prediction Module

The model adopts the vector inner product of $u_f$ and $v_f$ to measure the scores of user $u$ on item $v$:

$$\hat{y}(u, v) = u_f^T v_f \tag{25}$$

Finally, the complete loss function of the model is defined as:

$$L = \sum_{u \in U} \left( \sum_{v \in \{v | y_{uv} = 1\}} \Gamma(y_{uv}, \hat{y}_{uv}) - \sum_{v \in \{v | y_{uv} = 0\}} \Gamma(y_{uv}, \hat{y}_{uv}) \right) + \lambda_1 ||\Theta||_2^2 \tag{26}$$

where $\Gamma$ represents the cross-entropy loss function, $\theta$ represents the parameter to be trained, and $\lambda_1$ represents the hyperparameter used to control the $L2$ regularization.

## 4. Experiments

In this section, experiments are conducted on public datasets in the fields of film and music to verify the recommendation effect of the KGRFSF proposed in this paper. Datasets

are introduced first in Section 4.1; Section 4.2 introduces the experimental environment, hyperparameter setting, and experimental metrics. In Sections 4.3 and 4.4, the KGRFSF is compared with four other recommendation models based on a knowledge graph to verify the validity of the model. In Section 4.5, this paper first discusses the influence of the number of network layers contained in the two main modules of the model on the recommendation results; finally, the ablation experiment proves the effectiveness of the feature space fusion method proposed in this study.

*4.1. Datasets*

In the experiment, the public datasets of movie recommendations and music recommendations were used to test the recommendation effect of the model. Last.FM is derived from the Last.FM online music platform, and the dataset contains music interaction information of approximately 2000 users. MovieLens-20 M is one of the most widely used public datasets for movie recommendation scenarios, containing information about approximately 20 million users' movie ratings. The dataset was randomly separated into a training set, evaluation set, and testing set in accordance with the ratio of 6:2:2. The statistical results are shown in Table 1.

**Table 1.** Statistics of datasets.

| Datasets | Users | Items | Interactions | Entities | Relations | Triples |
|----------|-------|-------|--------------|----------|-----------|---------|
| Last.FM | 1872 | 3846 | 92,346 | 9366 | 60 | 15,518 |
| MovieLens-20 M | 138,159 | 16,954 | 13,501,622 | 102,569 | 32 | 499,474 |

*4.2. Parameter Settings*

4.2.1. Experimental Environment and Hyperparameter Setting

The experimental environment of this experiment is a Windows 64-bit operating system, NVIDIA GeForce GTX 1650, 16 GB. The experimental tools are PyCharm, Python 3.6, and deep learning PyTorch 1.0.

The hyperparameter settings of this experiment are shown in Table 2. In the table, $d$ is the dimension size of the embedding vector; $S_u$ and $S_v$ represent the size of the entity set related to the users and items knowledge graph. $\lambda$ represents the regularization coefficient of L2; Lr stands for learning rate; batch-size indicates the batch-size of batch training.

**Table 2.** Experimental parameter settings.

| Dataset | $d$ | $S_u$ | $S_v$ | $\lambda$ | Lr | Batch-Size |
|---------|-----|-------|-------|-----------|-----|------------|
| Last.FM | 64 | 8 | 16 | $1 \times 10^{-5}$ | $1 \times 10^{-5}$ | 1024 |
| MovieLens-20 M | 64 | 16 | 24 | $1 \times 10^{-5}$ | $1 \times 10^{-5}$ | 1024 |

4.2.2. Experimental Metrics

The AUC and F1 values served as evaluation metrics in the experiment. The calculation formula of the evaluation metrics is shown as follows.

AUC means the probability that the model predicts that the score of the user's favorite item is greater than that of the user's disliked item. The larger the AUC is, the better the model's prediction effect is. Further, where $m'$ is the number of times that the user's scores for their favorite items are greater than the user's scores for their disliked items, and $m''$ is the number of times that the user's scores for their favorite items are equal to the user's scores for their disliked items, $m$ is the total number of comparisons. The calculation of AUC is shown in formula (27).

$$AUC = \frac{m' + 0.5m''}{m} \tag{27}$$

Precision is the ratio of the intersection of the recommended item set of the model and the item set actually interacted with by the user in the recommended item set. The

higher the precision is, the larger the accuracy of the model recommendation, where *Fav(u)* represents the interactive item set of user U, and *Rec(u)* stands for the item set recommended to user *u* in the recommendation model proposed in this paper.

$$\text{Precision} = \frac{\sum_{u \in U} |Rec(u) \cap Fav(u)|}{\sum_{u \in U} Rec(u)} \tag{28}$$

Recall means the proportion of the intersection of item sets recommended by the model and item sets actually interacting with users to item sets actually interacting with users. The higher the recall is, the larger the recall rate of the model.

$$\text{Recall} = \frac{\sum_{u \in U} |Rec(u) \cap Fav(u)|}{\sum_{u \in U} Fav(u)} \tag{29}$$

F1 is the geometric average of accuracy and recall rate, and F1 can more comprehensively measure the effectiveness of the algorithm:

$$\text{F1} = \frac{2 \times \text{Precision} \times \text{Recall}}{\text{Precision} + \text{Recall}} \tag{30}$$

### 4.3. Baselines

To test the validity of the KGRFSF presented in this paper, this section will be compared with the following models:

- CKE [24] is a knowledge graph recommendation model based on embedding. CKE introduces auxiliary information, such as knowledge graphs and text, into a collaborative filtering algorithm for recommendation. The model is based on the TransR [25] algorithm to calculate the embedding vector of nodes in the knowledge graph, which serves to enrich the feature expressions of users and items.
- KGCN [7] extends graph convolutional networks to the field of KG recommendation. By combining the neighborhood information of the nodes of the knowledge graph, the higher-order association information between entities in the KG is mined to obtain richer representations of users and items.
- KGAT [8] integrates UIG and KG into CKG and makes an attention mechanism to obtain the neighbor information of users and items to obtain vector representations of users and items.
- CKAN [9] models the content feature of a knowledge graph based on an attention mechanism. The model constructs a set of content entities related to users and items through collaborative filtering propagation and combines the content feature contained in the set into the feature vector representations of users and items.

### 4.4. Performance Comparison

The F1 and AUC were used as evaluation metrics in the comparative experiment, and the experimental results are shown in Table 3. In the table, the optimal value of the evaluation index is highlighted in bold. Through looking at the experimental results, it can be found that the KGRFSF proposed in this section has obtained the optimal value in each index of the two datasets by comparison. Specifically, compared with the suboptimal value of each index, the AUC of KGRFSF on the Last.FM dataset improved by two percentage points, and F1 improved by three percentage points. The AUC on the MovieLens-20 M dataset improved by 0.6 percentage points, and F1 improved by 0.2 percentage points. Further analysis of the experimental results can obtain the following conclusions:

**Table 3.** Recommendation accuracy comparison experiment results.

| Model | Last.FM | | MovieLens-20 M | |
|---|---|---|---|---|
| | AUC | F1 | AUC | F1 |
| CKE | 0.747 | 0.674 | 0.927 | 0.874 |
| KGCN | 0.802 | 0.708 | 0.977 | 0.930 |
| KGAT | 0.829 | 0.742 | 0.975 | 0.929 |
| CKAN | 0.842 | 0.769 | 0.976 | 0.929 |
| KGRFSF | 0.862 | 0.799 | 0.982 | 0.931 |
| Improve/% | 2% | 3% | 0.6% | 0.2% |

Among them, the embedding knowledge graph recommendation model CKE has the worst experimental results because CKE only considers the first-order association information of nodes in the embedding calculation and cannot effectively excavate the higher-order association information in the user–item interaction graph and knowledge graph. However, KGCN and KGAT consider the high-order neighborhood information of project nodes in KG and the importance of different neighbor nodes in aggregation. However, the two models do not consider the high-order neighborhood information of users in KG. CKAN considers the user's interest path on the KG and obtains the user's interest vector but fails to make full use of the interactive behavior information contained in UIG. However, KGRFSF deeply excavates behavioral features and uses the feature space fusion method to fuse behavioral features and content features, enriching the vector expressions of users and items and reducing the impact of "noise" caused by simple fusion of behavioral features and content features in the same space, thus achieving better recommendation results.

### 4.5. Study of KGRFSF

#### 4.5.1. Influence of Network Layers

This section compares the influence of different network layers on the recommendation results of the behavioral feature module and content feature module through experiments.

First, the network layers of the content feature module are fixed as 1, and the network layers of the behavioral feature module are adjusted as 1, 2, 3, and 4. The experimental results are shown in Figure 4. The results show that the model obtains the best result when the behavioral feature module network has 3 layers and the content feature module network has 1 layer. This shows that fully mining the behavior information in the UIG is helpful to accurately model the behavioral features of users and items and enhance the model performance. When the network layers of behavioral feature modules are small, the model cannot fully extract behavioral features. However, if the selected neighborhood scope is too large, information with a low degree of association with users and items will be introduced, resulting in the decline in AUC on the Last.FM dataset and no further improvement in AUC on the MovieLens-20 M dataset.

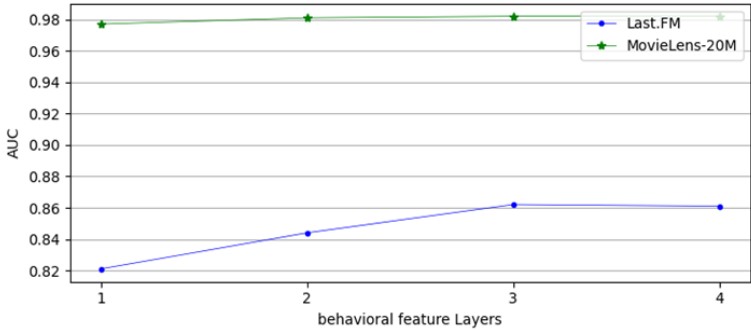

**Figure 4.** Influence of behavioral feature layers on AUC.

Next, the network layers of the content feature module are adjusted on the basis of the behavioral feature module network taking 3 layers, and the size of AUC is compared. The results are shown in Figure 5. By observing the experimental results, it can be found that AUC reduces gradually with the rise in network layers of content feature modules. The reason for this result is that the first-order neighbor in the knowledge graph can most accurately describe the content features of users and items, while the higher-order neighbor entity contains content features with low relevance to users and items, which interferes with recommendations. Therefore, when extracting content features in the knowledge graph, better results can be obtained by simplifying network layers.

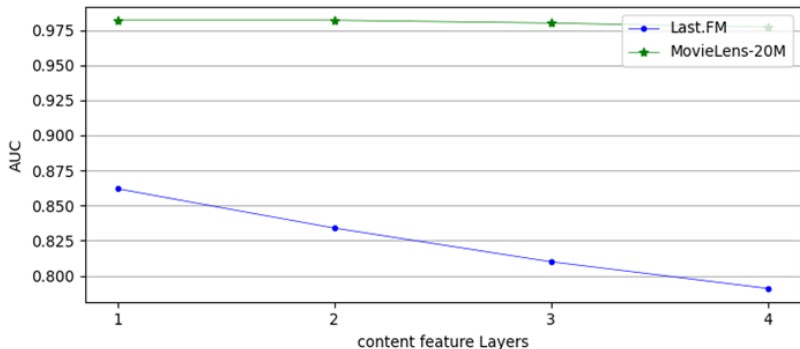

**Figure 5.** Influence of content feature layers on AUC.

In conclusion, aggregation of high-order neighborhood information is helpful to accurately obtain the behavioral features of users and items in the UIG. However, if the selected neighborhood scope is too large, information with a low correlation degree with the central node will be introduced. Experiments show that the model can obtain the best recommendation performance when the behavioral feature module network is at 3 layers. In the knowledge graph, first-order neighbors are most closely related to users and items, while higher-order neighbors may contain content irrelevant to users and items that interfere with recommendations. It can be found through the experimental results that the model can attain the best recommendation performance when the content feature module network is set to 1 layer.

### 4.5.2. Influence of the Fusion Method

This section verifies the effectiveness of the proposed feature space fusion method through comparative experiments.

In the experiment, the behavioral feature module and content feature module of KGRFSF remain unchanged. This section compares the feature space fusion method with other fusion methods:

- Sum aggregator: The behavioral feature and content feature are directly added and then fused by a multilayer perceptron.

$$agg_{sum} = \sigma(W_{sum}(e_{cf} + e_{kg}) + b_{sum}) \tag{31}$$

  where $W_{sum}$ and $b_{sum}$ are parameters to be learned. $e_{cf}$ and $e_{kg}$ represent behavioral features and content, respectively.

- Concat aggregator: The behavioral feature and content feature are spliced and then fused by a multilayer perceptron.

$$agg_{concat} = \sigma(W_{concat}(e_{cf}||e_{kg}) + b_{concat}) \tag{32}$$

  where $W_{concat}$ and $b_{concat}$ are parameters to be learned.

The other parameters of the experiment were set the same, and the experimental results are shown in Table 4. In the table, KGRFSF-$agg_{sum}$ uses the Sum aggregator, KGRFSF-$agg_{concat}$ uses the Concat aggregator, and KGRFSF-$agg_{project}$ uses the feature space fusion

method. The observation results show that KGRFSF-$agg_{project}$ has obtained the optimal values of all indicators in the two datasets. In addition, compared with the suboptimal results, the AUC and the F1 of the KGRFSF-$agg_{project}$ on the Last.FM dataset increased by 1.8 and 1.6 percentage points. The AUC and the F1 on the MovieLens-20 M dataset improved by 0.3 and 0.6 percentage points, respectively. The experimental results show that the presented feature space fusion method is able to effectively fuse behavioral features and content features and avoid the "noise" problem caused by simple fusion of the two. Finally, the accuracy of the recommendation results is improved.

**Table 4.** Influence of the fusion method on recommendation results.

| Fusion Method | | Last.FM | | MovieLens-20 M | |
|---|---|---|---|---|---|
| | | AUC | F1 | AUC | F1 |
| Other fusion methods | KGRFSF-$agg_{sum}$ | 0.844 | 0.779 | 0.974 | 0.924 |
| | KGRFSF-$agg_{concat}$ | 0.842 | 0.783 | 0.979 | 0.925 |
| Feature space fusion method | KGRFSF-$agg_{project}$ | 0.862 | 0.799 | 0.982 | 0.931 |

## 5. Conclusions

Here, this paper presents a recommendation model called the knowledge graph recommendation model based on feature space fusion (KGRFSF), which obtains behavioral features and content features in different feature spaces. The feature space fusion method enriches the feature vector representations of users and items and avoids the noise problem in the existing methods. Experimental results on Last.FM, a dataset published in music fields, and MovieLens-20 M prove the effectiveness of the model. However, the model holds that all interaction behaviors originate from the user's preference for items and does not consider that, in the real scene, there are diverse intentions behind the different interaction behaviors of users, ignoring the important role played by intention in the user's decision-making process, which is also our next research direction.

**Author Contributions:** Methodology, R.W.; Project administration, J.G. and J.L.; Writing—review & editing, S.Z. and X.W. All authors have read and agreed to the published version of the manuscript.

**Funding:** This work was supported by the Key research and development projects in Hebei Province (No. 20310802D).

**Institutional Review Board Statement:** Not applicable.

**Informed Consent Statement:** Not applicable.

**Data Availability Statement:** MovieLens-20M: https://grouplens.org/datasets/movielens/20m. Last.FM: https://grouplens.org/datasets/hetrec-2011 (all accessed on 28 June 2022).

**Conflicts of Interest:** The authors declare no conflict of interest.

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
