# Peer review of "Knowledge Graph Recommendation Model Based on Feature Space Fusion"

_applsci, doi:10.3390/app12178764_

Round 1

Reviewer 1 Report

1. Figure 1 is a critical component of this manuscript which is important for readers to understand the purpose of this study and the application of the proposed method.  Its current explanation is however inadequate, neither in the main text nor in the figure Legend.  Authors should at least explain the means of those colors, including those color-indicated lines and frames.  Readers won’t be able to effectively obtain a clear idea from the figure when they are looking at it and its caption.  Please improve it.

2. Probably authors can provide an additional introduction regarding the questions of sparse data and cold start to readers, especially to those who are not familiar with recommendation models.

3. Please elaborate the meaning of “additional noise” mentioned in row_103@page_3 with examples.

4. I have failed to understand the meaning of “Triples” mentioned in Table 1.

5. Please describe the numbers of Npos and Nneg and the ratio between them, mentioned in Eq. 27.

6. Some minors: (1) what is the “CF” mentioned in row_44@page_1?  (2) move the (KG) from row_48 to 47@page_2.  (3) typos “.” -> “,” in row_108@page_3 and row_332@page_10.  (4)

Reviewer 2 Report

Major changes are needed.

Reviewer 3 Report

The manuscript would benefit from an accurate reading. The novelty of the manuscript should be better presented.  Acronyms are used before they are introduced. 

Author Response

Thank you for the comments and helpful suggestions. We have taken the comments and suggestions into account, and have made corrections in the revised manuscript.

Point 1: The manuscript would benefit from an accurate reading. The novelty of the manuscript should be better presented. Acronyms are used before they are introduced.

Response 1:

Thank you very much for your advice. In order to better describe our method, clearly reflect the novelty of the manuscript, and facilitate readers to read, we have made the following modifications to the manuscript.

First, a theoretical analysis of the causes of noise generated by the model is added in lines 119-223. At the same time, in Section 3.1, we explain the causes of noise in detail through examples.

Then, in order to solve the noise problem mentioned above, we have added a description of the application of the projection method in Section 3.2.3 (feature space fusion module) of the manuscript. Meanwhile, in the fourth section, we have shown in Table 3.4 by adding a row of improved percentage data. It is intuitively demonstrated that the proposed knowledge graph recommendation model based on feature space fusion can effectively improve the recommendation accuracy compared with the existing methods.

Finally, we've done a thorough review of abbreviations. We have added the acronym CF where we first mentioned collaborative filtering. At the same time, we move the acronym KG in line 48 to the place where the knowledge graph first appears.

Reviewer 4 Report

This paper introduces a new recommandation algorithm based on user-item interactions, called the the knowledge graph recommendation model based
on feature space fusion (KGRFSF). First, the introduction and the state of the art part describe existing recommandation models and their limits, then part 3 describes the structure of the KGRFSF model and how each phase of the model works, and part 4 applies this new model to two datasets and shows improvements relative to existing models.

The article is generally well-written, the proposed method is complex but, for the most part, every stage is explained clearly, the experiment design is described well, and the experiment in itself adequately shows the interest and the performance of the method, even if the improvements are relatively modest. The level of writing is generally very high, apart from a few sentences that are not grammatically sound and should be rewritten. There also seem to be several errors in the mathematical expressions in different equations, but they do not affect the understanding of the article too much.

My main demand to improve the article is in part 3.2.3 (Feature Space Fusion Module). While the other parts of the method are adequately explained, the projection is not sufficiently explained for me. The authors should explain more how the projection space is structured, how behavioural feature vectors and content features vectors are projected into it, and how its structure enables to filter irrelevant information while keeping the most important features. These explanations are all the more needed that the authors state that projecting both the behaviour features and the content features in the same space before the fusion is the main innovation of the paper, and that it allows the fusion of structures containing different kinds of information while filtering spurious or useless information. In part 4, the authors also show that this particular fusion procedure (involving a projection on a common space) is key in the performance of their method.

As some of the improvements in Part 4 look modest, it would be useful for the author to discuss the size of these improvements to support their conclusions.

Here are some remarks on the mathematical expressions that I believe are not completely correct:

1- Following equations (3), (4) and (5), I think there is probably an error in equation (6). As I understand it, the vector E^{(k)} corresponds to vectors u^{(k}) and v^{(k)}, so there is no need to multiply them by the matrices \left( D^{-1/2}AD^{-1/2} \right) ^ k as they are already the results of the multiplication of E^{(0)} by these matrices. As a result, equation (6) should be

E= \alpha_{0} E^{(0)} + \alpha_{1} E^{(1)} + \alpha_{2}E^{(2)}…

or

E=\alpha_{0} E^{(0)} + \alpha_{1} \left( D^{-1/2}AD^{-1/2} \right) E^{(0)} + \alpha_{2} \left( D^{-1/2}AD^{-1/2} \right) ^ 2 E^{(0)}…

2- l.238, the authors state that the item’s initial set of content entities consists of the item itself, so equation (8) should be \varepsilon_{v}=\{v\}. as it is written (i.e. \varepsilon_{v}=\{ v \mid v\in G \}, it is confusing because v acts like both an implicit and an explicit variable, which could lead to understand that \varepsilon_{v}=G, which is not the intention of the authors.

3- Equations (9) and (10) are not clear because r is not defined beforehand (we understand later that it is a relation between h and t. Besides, as h is used as an implicit variable in the definition of

\varepsilon_{u}^{(l)} and \varepsilon_{v}^{(l)}, it should be written as h in all occurrences, and h₁ and h₂ should be replaced by h.

Finally, here is a list of mostly grammatical or typographical errors, and a few

l.71 : I would reformulate : the information contained in the knowledge graph cannot cover all aspects of the item

l.72 : I do not understand what is meant by : ‘and the user interaction is not only for item content preferences’ ; does it mean that user interaction contains useful information that goes beyond item preference ?

l.108 : there is a useless period between ‘start’ and ‘faced’

l.192 : ‘Users to watch a movie’ seems superfluous and does not make sense grammatically ; the sentence would be correct without this (starting with ‘for example’)

l.195-196 : ‘friend recommandation’ appears twice in the enumeration

l.245, the sentence should begin with ‘We obtain the entity...’, for example.

l.261-263 are very hard to understand ; the readers expects such a sentence to have the structure :

‘where the attention network […] is […], the last layer of activation […] is […], etc., but we only have this structure to explain what | |, W, b and r’ are (and as a result the end of the sentence is easier to understand).

l.277, the beginning of the sentence ‘Model for solving the question’ is not grammatically correct ; the beginning of the sentence could be : ‘Among the models that could solve the question, ...’

l.279-280, I do not understand what ‘projection to the preference’ means ; the whole sentence is probably incorrect grammatically and should be rewritten.

l.291-294 : the hyperparameter is noted \lambda_{1} in equation (26) and \lambda in the text (line 294)

l.291-293 : sets $P^+$ and $P^-$ are described in the text but not used in equation (26)

l.338-339 : Rec(u) is the item set recommanded to the user (not by the user)

l.372 : ‘Last. The FM dataset’ should be ‘the Last. FM dataset’
